# Distribution Trends of Cadmium and Lead in Timberline Coniferous Forests in the Eastern Tibetan Plateau

**Longyu Jia [1], Ji Luo [2], Peihao Peng [1],\*, Wei Li [2], Danli Yang [1], Wenbo Shi [1], Qian Xu [1] and Xiyi Lai [1]**

[1] College of Tourism and Urban-Rural Planning, Chengdu University of Technology, Chengdu 610059, China; jialongyu@stu.cdut.edu.cn (L.J.); yangdanli@stu.cdut.edu.cn (D.Y.); shiwenbo@stu.cdut.edu.cn (W.S.); xuqiann@stu.cdut.edu.cn (Q.X.); laixiyi@stu.cdut.edu.cn (X.L.)

[2] Key Laboratory of Mountain Surface Processes and Ecological Regulation, Institute of Mountain Hazards and Environment, Chinese Academy of Sciences, Chengdu 610041, China; luoji@imde.ac.cn (J.L.); liwei@imde.ac.cn (W.L.)

\* Correspondence: pengpeihao@cdut.edu.cn

**Abstract:** The concentrations of Pb and Cd in the needles and twigs of fir and spruce collected from 26 sites in the Eastern Tibetan Plateau were measured and analyzed in this study. The mean concentrations of Cd and Pb were 0.034 and 1.291 mg/kg, respectively, in the needles and 0.101 and 2.511 mg/kg, respectively, in the twigs. These concentrations increased significantly with needle and twig age and peaked at 5 years. The twigs were significantly enriched in Pb and Cd compared with the needles. The spatial distributions of Pb and Cd were determined using the inverse-distance-weighted spatial interpolation method on the basis of the mean concentration of the elements in the needles and twigs from each site. The highest concentrations of Pb and Cd in twigs and needles were found in Yunnan Province and Gongga Mountain. They showed a tendency to decline from Yunnan Province to the northern direction, as well as from Gongga Mountain to the western area. Principal component analysis showed that Pb and Cd originated from the anthropogenic activities in this area. The mining activities and climatic factors may be the main sources of Pb and Cd in this area. Combining the HYSPLIT (The Hybrid Single-Particle Lagrangian Integrated Trajectory) model and PCA, the results implied that exterior Pb and Cd sources from Southeast Asia and the eastern developed cities in China can infiltrate the ecosystem through long-range transportation and accumulate in timberline forests, with atmospheric deposition in the Eastern Tibetan Plateau. This plateau suffers from severe Pb pollution but slight Cd contamination.

**Keywords:** cadmium; lead; distribution trends; timberline forest; bio-monitor; anthropogenic source

## 1. Introduction

Pb and Cd are widespread in the earth's crust. Enormous amounts of Pb and Cd are emitted into the atmosphere with urbanization, rising human population, and excessive industrialization, causing severe environmental pollution [1]. Some studies have also reported Pb and Cd contamination among children in China, who presented with hair loss and dysplasia due to high blood Pb levels [2]. Increased blood lead levels among people have also been revealed in other countries [3,4]. The two elements can deposit in forests via wet and dry deposition [5], impacting the balance of ecosystems [6]. Forest canopies retain high amounts of Pb and Cd because they consist of a large active surface interacting with these pollutants [7]. Although unknown biological roles are displayed in plants, high accumulation of Pb and Cd produces severe toxicity, inhibits the growth of plants, and even causes plant death [8,9].

Spermatophytes have been regularly applied to monitor the pollution of heavy metals [10]. Many studies have reported that the heavy metal concentration in leaves is significantly correlated with ambient pollutant content, with no significant correlation between foliage and the concentrations in soil [11], implying that foliage is fit for the bio-monitoring of atmospheric pollution [12]. Twigs have been proved as important indicators

of heavy metal pollution [13], suggesting that twigs can also be used to indicate heavy metal pollution [14].

Many previous studies have focused on the vicinity of polluted regions to metallurgical industry, steel works, and urban areas [15,16]. Little attention has been paid to the timberline forests in the Tibetan Plateau, which is remote from pollutants but sensitive to their infiltration. The average elevation of the study area is 3800 m above sea level, with rare anthropogenic activities and pollutant sources. However, infiltration of Pb and Cd has been reported in the eastern Tibetan Plateau [17]. Pb and Cd enter into the atmosphere in the Tibetan Plateau from northeastern India with long-distance transportation [18], affecting primeval ecosystems [19]. Because numerous rivers with important ecological functions originate in this area, continuous and immense input of pollution will result in serious risks downstream, affecting ecosystem stability [20]. It is important to monitor trace metal concentrations in the timberline forests in the eastern Tibetan Plateau. Spruce was selected as a bio-monitor of Pb and Cd because it is a widespread and typical species in the area. Coniferous needles were able to assimilate trace metals from the atmosphere, which have been proved as indicators of air pollution [12,15,21,22].

The present study was conducted to: (1) evaluate the levels of Pb and Cd in spruce in the timberline forests in the Tibetan Plateau; (2) discuss the possible sources of Cd and Pb and their influence factors in needles and twigs.

## 2. Materials and Methods

### 2.1. Site Description

The study area is located in the timberline forests in the middle of the Hengduan Mountains, eastern Tibetan Plateau, China. It is a junction of the western Sichuan Province, the northern Yunnan Province, and the eastern Tibetan Autonomous Region of China. The Hengduan Mountains consist of a series of mountain ranges and rivers stretching from north to south. Their altitude ranges from 4000 to 5000 m above sea level. Most rivers empty into the Pacifica Ocean, except for the Nujiang River, which drains into the Indian Ocean. The Hengduan Mountains are controlled by the westerlies, the southwest monsoon from the Indian Ocean, and the southeasterly monsoon from the Pacific Ocean, causing a wet season (from the middle of May to the middle of October) and a dry season (from the middle of October to the middle of May next year) in this region. The wet season is characterized by tremendous rainfall with elevated temperature and humidity, and the dry season by rare precipitation, long sunlight, huge evaporation, and dry atmosphere. The remarkable species and the biodiversity of the region are attributed to its unique climatic factors and topographic features. Because of the influence of the steep mountains and chip movement in the rivers, the topography is magnificent, with gorges and ranges. The topography shows a decrease in elevation from northwest to southeast, as well as a decline in precipitation. The scenery of the mountains varies over the year, and the landscape differs dramatically between foot and mountaintop.

### 2.2. Sampling

The needles and twigs were collected from 26 sampling sites (Figure 1). At each sample point, 20 × 30 m sample plots were established with three replicates. Each sample plot comprised twenty-four 5 × 5 m quadrants [23]. Twelve quadrants in each sample plot were randomly picked for sampling. The needles and twigs were collected from one tree (spruce) at the same branches in each quadrant at an average height of 2 m above ground from all directions. The samples were kept in neat cellulose bags and stored under cold conditions. When brought to the laboratory, these samples were washed with distilled water to remove adhering particles on the needle and twig surfaces. Then all samples were oven-dried to a constant weight at 60 °C for 24 h and ground to pass through a 0.2 mm screen. All samples were kept in contamination-free polyethylene plastic bags until chemically analyzed.

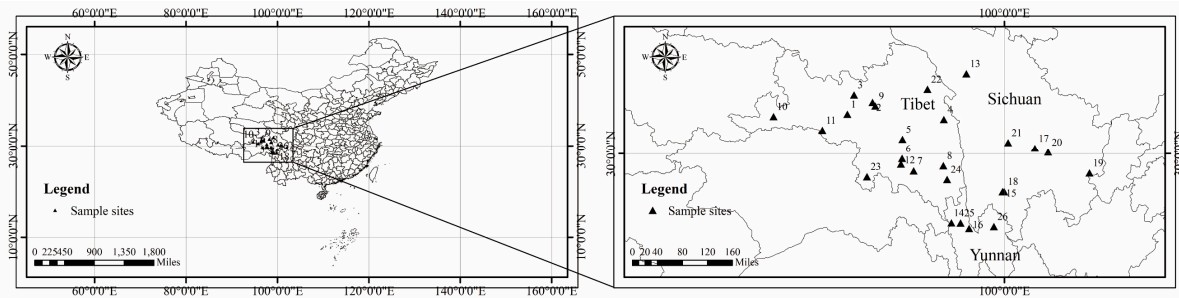

**Figure 1.** Sampling sites of conifer needles and twigs (fir and spruce) in the eastern Tibetan Plateau, China.

### 2.3. Element Analysis

In research, it is important to accurately detect the concentration of heavy metals. Multiple methods are used to determine the concentrations of heavy metals. Atomic absorption spectrometry (AAS) has been widely used for its simple operation and good accuracy [24]. However, it requires expensive instrumentation and provides inefficient measurements. Electrochemical methods have received interest for their quick detection speed, easy manipulation, low cost, and high sensitivity [25]. However, due to their serious toxicity and difficult handling, an iron oxide/graphene composite, with moderate toxicity and stability, has drawn rising attention to replace it [26]. Plasmonic optical fiber gratings functionalized with bacteria are a new, interesting technique to detect heavy metals with high sensitivity [27]. These methods cannot provide simultaneous detection when the species of elements are diverse. In the present study, inductively coupled plasma mass spectrometry (7700×, Agilent Technologies, Palo Alto, CA, USA) was applied to detect heavy metals because of its optimum sensitivity, highest accuracy, and effective speed [28,29]. We used it to analyze solutions acidified with microwave-assisted nitric acid, hydrogen peroxide, and hydrofluoric acid. Pb and Cd were detected using ICPMS, following the USA EPA 6020a method (Revision 1, February 2007). The detection limits of Pb and Cd were 0.001 and 0.002 µg/L, respectively. Quality control was ensured through the analysis of two reference materials from the National Quality and Technology Supervision Agency of China (GBW07603 and GBW07604). Accuracy was checked by measuring reference materials. The measurement errors were below 5%, and the recovery of trace metals ranged from 99% and 110% of the reference materials. Geostatistical analysis was performed by ArcGIS 10.6. Descriptive statistics of Pb and Cd between needles and twigs were carried out by SPSS version 12.0 software with the Mann–Whitney U test. The significance level was set as $p < 0.05$.

### 3. Result

#### 3.1. Mean Concentrations of Pb and Cd in Needles and Twigs

The analytical results (minimum, maximum, and mean values, and SD) of Cd and Pb in the needles and twigs are presented in Table 1. The mean concentrations of Cd and Pb were 0.034 and 1.291 mg/kg in the needles and 0.101 and 2.551 mg/kg in the twigs, respectively. Whether in needles or twigs, the average concentration of Pb was much higher than that of Cd. For both Pb or Cd, their mean content in the twigs was significantly greater than that in the needles.

**Table 1.** Concentrations of heavy metals in needles and twigs ($n = 28$).

| | Heavy Element Concentrations (mg kg$^{-1}$) | | | | | | | | |
|---|---|---|---|---|---|---|---|---|---|
| | Needles ($n = 30$) | | | | | Twigs ($n = 19$) | | | |
| | Min | Max | Mean | SD | | Min | Max | Mean | SD |
| Cd | 0.003 | 0.167 | 0.034 | 0.070 | Cd | 0.021 | 0.388 | 0.101 | 0.084 |
| Pb | 0.037 | 10.576 | 1.291 | 2.371 | Pb | 0.102 | 14.311 | 2.551 | 3.711 |

*3.2. Spatial Distribution Maps of Pb and Cd in Needles and Twigs*

The spatial distributions of Pb and Cd are presented with the inverse-distance-weighted spatial interpolation method on the basis of the mean concentrations of the elements in the needles and twigs from each site in Figure 2. High concentrations of Pb in the needles and twigs, denoted by red and orange on the maps, respectively, were found in the southeastern and east regions, while diluted concentrations were observed in the western direction. High Cd concentrations in the needles were found in the southeastern, eastern, and northern regions of the study area, while the southeastern and eastern regions were characterized by elevated concentrations of Cd in the twigs. Regardless of the heavy metal type in needles or twigs, its concentration in the western region was relatively lower.

## 4. Discussion

*4.1. Concentration of Pb and Cd in Needles and Twigs*

In the present study, a comparative investigation of Pb and Cd contents with the results for the needles allowed a more profound characterization of the current pollution situation (Table 2). Our results for Pb and Cd were similar to those of some reports on the eastern Tibetan Plateau [30]. Pb concentrations in needles in the present study were much higher than those measured in remote Austrian forests and even exceeded those in areas close to pollutants [15], whereas Cd concentrations were lower than those determined in remote sites in Austria [15] and France [31]. This might imply that the eastern Tibetan Plateau suffers from severe Pb pollution and the content of Cd is normal. This is consistent with the findings reported by Tang et al. (2014), who found that Cd concentrations in soil were no more than the national standard while Pb concentrations were higher than the national standard in the eastern Tibetan Plateau. The lead content was determined in modern groundwater in the Tibetan Plateau, showing that natural sources account for only a small portion, indicating that the Tibetan Plateau has been polluted by human activities [32]. The analysis of the Pb concentration in the river water and ice core in the Tibetan Plateau also demonstrated that the Qinghai-Tibet Plateau has been polluted to some extent [33,34]. Increasing amounts of evidence have suggested that pronounced Pb pollution has entered into the Tibetan Plateau, with an increasing trend in the past decades. Because of growing demands for energy in the region and increasing industrial production, the Asian emissions are the largest compared with other continents, presenting a rising tendency. However, the emission of trace metals has decreased over the last decades in Europe and North America due to a series of environmental protection measures [35]. More attention should be paid to the sensitive Tibetan Plateau. The behavior of Cd and Pb on the Tibetan Plateau plays a key role in the global biogeochemical cycle of Cd and Pb [36].

**Table 2.** The mean concentrations of Pb and Cd in other needles.

| | | | Concentrations (mg/kg) | | |
|---|---|---|---|---|---|
| | *n* | Site | Pb | Cd | |
| 1 | 126 | Tibetan Plateau | 0.92 | 0.05 | Tang et al. (2014) |
| 2 | 50 | Remote from a source in Norway | 0.4 | 0.04 | Trimbacher and Weiss (2001) |
| 3 | 25 | Steel works in Norway | 0.6 | 0.06 | Trimbacher and Weiss (2001) |
| 4 | 27 | Metallurgical industry in Norway | 0.8 | 0.11 | Trimbacher and Weiss (2001) |
| 5 | 11 | Highways in Norway | 1.1 | 0.08 | Trimbacher and Weiss (2001) |
| 6 | 11 | South France | 0.2 | 0.09 | L. Gandois and A. Probst (2012) |
| 7 | 48 | Lithuania | 0.77 | 0.09 | D. Ceburnis, E. Steinnes (1999) |
| 8 | 7 | Southern Norway | 0.89 | 0.173 | Berthelsen et al. (1995) |
| 9 | 7 | Central Norway | 0.03 | 0.011 | Berthelsen et al. (1995) |
| 10 | 26 | Tibet Plateau | 1.291 | 0.034 | Present study |

*4.2. Comparison of Pb and Cd between Needles and Twigs*

Table 3 shows that the Cd and Pb contents in the twigs were about three- and two-fold higher than in the needles, respectively. The concentrations of both Cd and Pb in the twigs

were significantly higher compared with the needles. It was reported that Cd and Pb accumulate in the twigs and needles with rising age. However, we were uncertain about the age of the collected needles and twigs. Therefore, we did not know whether organ or age difference caused the higher concentration in the twigs. Because of the difficulty of collecting needles and twigs of different ages from every site, the obstacle of distinguishing the age, and heavy workload, only twigs and needles whose age ranged from one to six years were collected from Gongga Mountain to explore the influence of age. Figure 3 shows that Pb and Cd concentrations between both organs increased with rising age and peaked at 5 years. Accumulation with increasing age in needles and twigs was confirmed. Even the highest concentrations of Pb and Cd in the needles were lower than those in the twigs, demonstrating that organ difference attributed to the content variation of Pb and Cd, with little contribution of age. The delivery from root to needle and the transfer from needle to root were the two main sources of heavy metals detected in plants [37,38]. In the present study, we observed higher contents of Pb and Cd in twigs. We found that Pb and Cd are mainly transferred from twigs to needles. It was claimed that toxic elements can transport to non-essential parts. The twigs might be a hub for the transfer of elements between organs [20]. The transportation between twigs and needles was bilateral. Comparing twigs to needles, Pb and Cd were diluted from needles to twigs. Different ages of needles and twigs from more sites are needed to better understand the behavior of Pb and Cd between twigs and needles for further study.

**Table 3.** The results of statistical analysis (Mann–Whitney U test).

| Parameters | Mann–Whitney U | Wilcoxon W | Z | $p$ | $n$ | Twigs/Needles |
|:---:|:---:|:---:|:---:|:---:|:---:|:---:|
| Cd | 144 | 495 | −3.511 | $p < 0.001$ | 26 | 1.9 |
| Pb | 50 | 199 | −2.544 | $p < 0.05$ | 26 | 2.9 |

*4.3. Sources of Pb and Cd between Needles and Twigs*

Principal component analysis and factor analysis were performed to reveal the relationship of Cd and Pb with other trace elements and potential sources between needles and twigs in the eastern Tibetan Plateau. The initial component matrix displayed vague results. The matrix was rotated to eliminate ambiguities. The results suggested that the concentrations of heavy metals can be divided into three components, which account for almost 71% of the total variance in the data (Table 4). V, Cr, Co, Ni, Cu, and As have more than 34% factor loadings in the first principal component (PC1). V, Cr, Co, Ni, Cu, and As primarily originate from the crust in background organic soils [39]. Therefore, we assumed that PC1 represents natural sources.

PC2 consisted of As, Sb, Pb, Cd, and Hg, all of which belong to the group of five toxic elements with severe toxicity. PC2 had up to 0.713 and 0.465 factor loadings for Pb and Cd, respectively. Pb and Hg are elements of the long-range transport family [18]. Cd can be transported over long distances via the atmosphere [40]. We speculated that PC2 represented climatic factors. Figure 2 shows a tendency to decline for elements in needles and twigs from Yunnan Province in the northern direction as well as from Gongga Mountain to western areas. The regular distribution patterns implied that climatic factors are more dominant compared with mining activities. Combining PCA results and distribution patterns, a definite conclusion can be drawn: climatic factors overtly affect the distribution of Cd and Pb in needles and twigs. The hybrid single-particle Lagrangian integrated trajectory (HYSPLIT) model of 72 h backward trajectories was performed to explore the possible sources of aerosol and transport routes of air masses. The trajectories were computed for an altitude of 3800 m in 2012. Figure 4 presents the results for each month. From January to May and October to December, the investigated region was affected by the atmosphere, mainly originating from northern and northeastern India. In June, the air mass mostly came from northwestern Yunnan. The atmosphere from South Asia and developed cities located in eastern China had an impact on the Tibetan Plateau.

Severe air pollution was reported in South Asia [41,42]. The atmospheric mass mostly came from developed cities in eastern China and southwestern Yunnan Province during summer [18]. Figure 2 shows that pronounced concentrations of Cd and Pb were found in the south and east directions, which were influenced by southwesterly and southeasterly monsoon. Massive amounts of Pb and Cd particles produced by mining activities, steel works, and metallurgical industry were delivered to the Tibetan Plateau through long-distance transportation via wide ranges [43] as well as dry and wet deposition [18,44].

**Table 4.** Component matrix.

| Element | Component Matrix | | | Rotated Component Matrix | | |
|---|---|---|---|---|---|---|
| | 1(42%) | 2(16%) | 3(12%) | 1(35%) | 2(19%) | 3(17%) |
| V | 0.885 | −0.233 | −0.175 | 0.903 | 0.186 | 0.136 |
| Cr | 0.775 | −0.298 | 0.368 | 0.701 | −0.105 | 0.567 |
| Co | 0.902 | −0.105 | −0.171 | 0.857 | 0.3 | 0.172 |
| Ni | 0.499 | −0.475 | 0.018 | 0.64 | −0.233 | 0.109 |
| Cu | 0.682 | −0.406 | −0.339 | 0.853 | 0.021 | −0.126 |
| Zn | 0.63 | 0.048 | 0.116 | 0.481 | 0.23 | 0.358 |
| As | 0.926 | 0.006 | −0.228 | 0.84 | 0.425 | 0.155 |
| Mo | 0.413 | 0.073 | 0.841 | 0.101 | −0.087 | 0.93 |
| Sb | 0.424 | 0.588 | −0.24 | 0.147 | 0.746 | 0.072 |
| Pb | 0.591 | 0.661 | 0.199 | 0.141 | 0.713 | 0.546 |
| Cd | 0.424 | 0.474 | 0.272 | 0.068 | 0.465 | 0.508 |
| Hg | 0.222 | 0.621 | −0.426 | 0.008 | 0.767 | −0.166 |

The title row spans: **Component Matrix (70.642% of Total Variance Explained)**

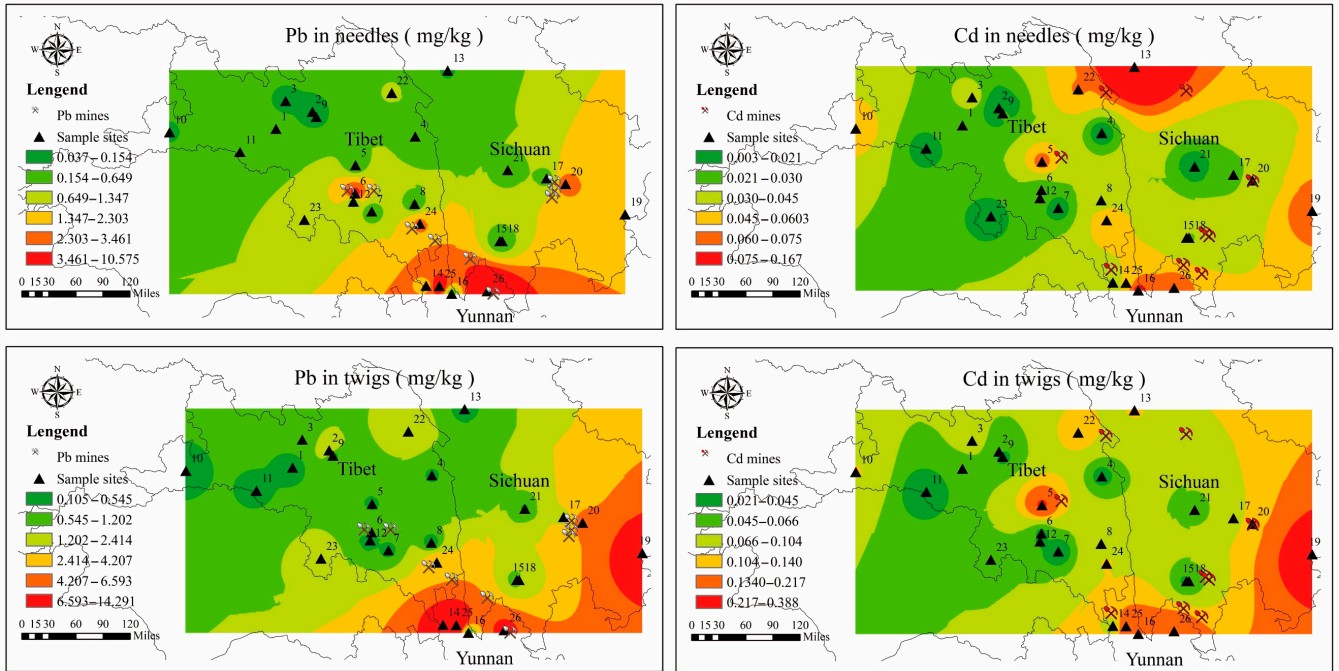

**Figure 2.** Spatial distribution map of Pb and Cd in the needles and twigs in the mining area close to our study area (https://www.sohu.com/a/365087860_660942). The red and orange colors on each small map represent relatively high concentration of Pb and Cd, while the green and lime colors represent relatively low concentrations, respectively. The buff color represents the middle values.

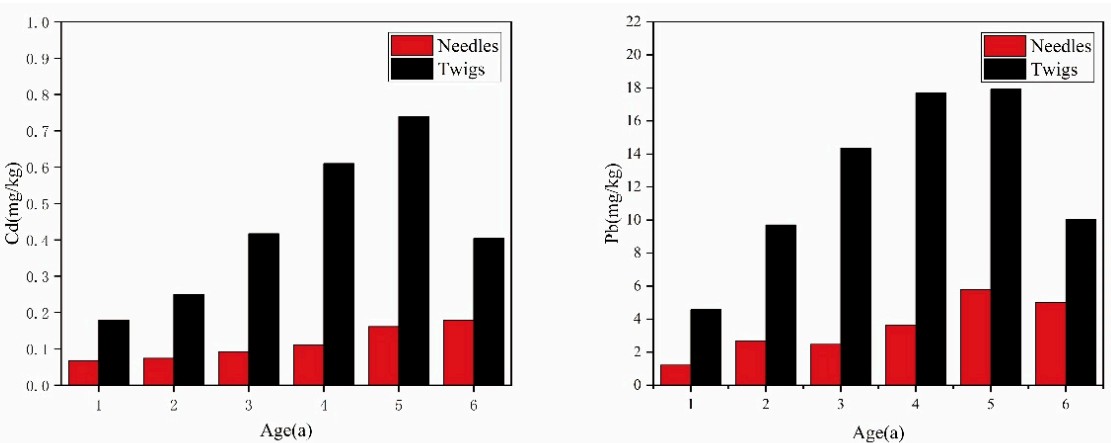

**Figure 3.** Pb and Cd in twigs and needles of different ages.

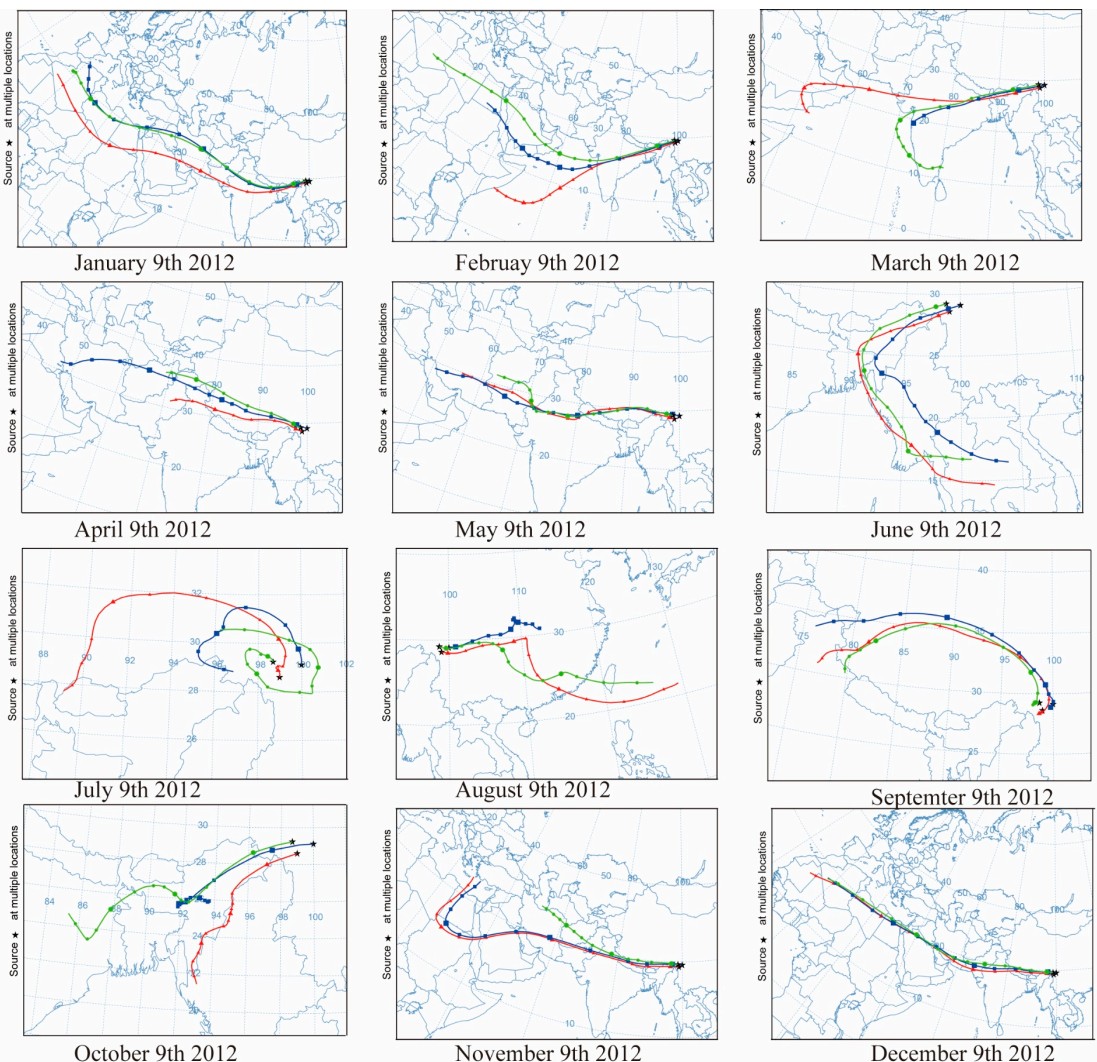

**Figure 4.** The 72 h backward trajectories of the sampling period of the study area from January to December 2012 (https://ready.arl.noaa.gov/hypub-bin/trajtype.pl?runtype=archive).

Pb and Cd had high factor loadings in PC3, which might originate from local sources. To investigate probable anthropogenic sources of Cd and Pb in needles and twigs in detail, the distribution patterns of lead and cadmium mines in the vicinity of the study area

were analyzed. Figure 2 shows that relatively high concentrations of Cd were near the mines of Cd, and the highest concentrations of Pb were found in the vicinity of large Pb mines. Mines, an important factor influencing Cd and Pb concentrations in needles and twigs, partly accounted for the distributions of Pb and Cd. However, sites with relatively high concentrations far away from the mines were also observed, which indicated that other factors exist besides mines. Heavy metals are widely used in industrial processes and mining enterprises, which directly discharge wastewater without strict treatment, making the soil around them susceptible to high levels of toxic heavy metals. Pb was always associated with emissions from car traffic [45]. Cd concentrations were elevated near highways [15]. These reports suggested traffic plays an important role for Cd and Pb concentration in needles. Previous literature showed that local sources of heavy metals severely affect mountain forest ecosystems [46]. The results indicated that anthropogenic activities mainly account for lead and cadmium in the eastern Tibet Plateau.

## 5. Conclusions

In the present study, the average concentrations of Cd and Pb were 0.03 and 1.2 mg/kg, respectively, in the needles and 0.101 and 2.551 mg/kg, respectively, in the twigs. In general, the Pb and Cd concentrations in the twigs were higher than that in needles. Anthropogenic activities were the main contributors of Cd and Pb. Mines and climatic factors played critical roles in their distribution patterns. Large quantities of the pollutants Cd and Pb were carried in the air mass from India and southeast monsoons. Higher Pb and Cd concentrations in needles and twigs were observed in Yunnan Province and Gongga Mountain. Considerable Cd and Pb infiltrated into the Tibetan Plateau. Pb pollution was severe, while that of Cd was slight. To better comprehend the characteristics of Cd and Pb in timberline forests, their concentrations and the storage of root, bark, litter, xylem, and soil should be considered. More sites should also be added.

**Author Contributions:** Conceptualization, L.J., P.P. and J.L.; methodology, L.J., P.P. and J.L.; formal analysis and investigation, L.J., P.P., J.L., D.Y., W.L., X.L., W.S. and Q.X.; writing—original draft preparation, L.J.; writing—review and editing, L.J.; resources: L.J., J.L., D.Y., W.L., X.L., W.S. and Q.X.; supervision, J.L. and P.P. All authors have read and agreed to the published version of the manuscript.

**Funding:** This study was funded by No. 2019QZKK0307, No. 41771062, and No. 2016YFC0503305.

**Institutional Review Board Statement:** Not applicable.

**Informed Consent Statement:** Not applicable.

**Data Availability Statement:** The data presented in this study are available on request from the corresponding author. The data are not publicly available due to the preciousness and confidentiality of data.

**Acknowledgments:** This research was financially supported by the Second Tibetan Plateau Scientific Expedition and Research Program (STEP) (Grant No. 2019QZKK0307), the National Natural Science Foundation of China (Grant No. 41771062) and the National Key Research and Development Program of China (Grant No. 2016YFC0503305).

**Conflicts of Interest:** The authors declare no conflict of interest.

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
