# Peer review of "Distribution Trends of Cadmium and Lead in Timberline Coniferous Forests in the Eastern Tibetan Plateau"

_applsci, doi:10.3390/app11020753_

Round 1

Reviewer 1 Report

This manuscript studies the distribution of Cd and Pb concentrations in the Tibetan plateau, a certain region of China. This is a very interesting work for the research community and the environmental preservation agencies, since the presence of heavy metals in nature significantly affects the whole ecosystem, with lots of negative consequences. In the text, the authors analyze the Cd and Pb levels of needles and twigs collected from different regions, they compare them and they establish the distribution maps of the two heavy metals. Finally they introduce several conclusions pointing out the origins of the heavy metals, mainly caused by anthropological causes. The paper is very well organized, providing proper explanations and clear graphs showcasing the results. The English usage is mostly correct as well. All in all, the manuscript is a good match for Applied Sciences. However, I will require the authors to perform a couple of minor but mandatory changes before considering the paper for publication:

- In the Introduction, the authors mention the negative consequences of heavy metals for the ecosystem, and for the plants in particular. However, they should not forget that heavy metals have numerous negative effects on humans as well. Heavy metals present in plants used in human nutrition can be transmitted to humans through ingestion. The authors should mention that heavy metals do not affect just plants, but also crops and therefore humans as well. The authors should point out as well some of the negative effects of heavy metals in humans.

- In section 2.3, the authors mention the technique they use to analyze the presence of heavy metals in needles and twigs. They should mention as well that research on heavy metals detection is of utmost importance nowadays and that other interesting techniques are arising, providing a plethora of mechanisms for heavy metals sensing. Please, include the following two references of heavy metals sensors as well: Talanta 160, 528-536 (2016) & Optics Express 28(13), 19740-19749 (2020).

Reviewer 2 Report

The manuscript entitled "Distribution trends of cadmium and lead in
timberline coniferous forests, eastern of the Tibet Plateau "is a good work, well written in all its parts.
The topic of pollution is very topical, and the authors treat it adequately.
I wonder why the authors did not think of monitoring the soil and water to evaluate the accumulation of metals, it would have given more heaviness to the manuscript as was done in a work "Toxic inorganic pollutants in foods from agricultural producing areas of Southern Italy: Level and risk assessment "Ecotoxicology and Environmental Safety 148, 2018, Pages 114-124.
